# Assessing the Reliability of Global Carbon Flux Dataset Compared to Existing Datasets and Their Spatiotemporal Characteristics

**Zili Xiong** [1], **Wei Shangguan** [1,*], **Vahid Nourani** [2,3], **Qingliang Li** [4], **Xingjie Lu** [1], **Lu Li** [1], **Feini Huang** [1], **Ye Zhang** [1], **Wenye Sun** [1], **Hua Yuan** [1] and **Xueyan Li** [5]

1 Southern Marine Science and Engineering Guangdong Laboratory (Zhuhai), Guangdong Province Key Laboratory for Climate Change and Natural Disaster Studies, School of Atmospheric Sciences, Sun Yat-sen University, Guangzhou 510275, China; xiongzli@mail2.sysu.edu.cn (Z.X.); luxingj@mail.sysu.edu.cn (X.L.); lilu35@mail2.sysu.edu.cn (L.L.); huangfn3@mail2.sysu.edu.cn (F.H.); zhangy929@mail2.sysu.edu.cn (Y.Z.); sunwy23@mail2.sysu.edu.cn (W.S.); yuanh25@mail.sysu.edu.cn (H.Y.)

2 Center of Excellence in Hydroinformatics, Faculty of Civil Engineering, University of Tabriz, Tabriz 51368, Iran; nourani@tabrizu.ac.ir

3 Faculty of Civil and Environmental Engineering, Near East University, Via Mersin 10, Nicosia 99628, Turkey

4 College of Computer Science and Technology, Changchun Normal University, Changchun 130032, China; liqingliang@ccsfu.edu.cn

5 Guangdong Provincial Key Laboratory of Remote Sensing and Geographical Information System, Guangdong Open Laboratory of Geospatial Information Technology and Application, Guangzhou Institute of Geography, Guangdong Academy of Science, Guangzhou 510070, China; lixueyan@gdas.ac.cn

* Correspondence: shgwei@mail.sysu.edu.cn

**Abstract:** Land carbon fluxes play a critical role in ecosystems, and acquiring a comprehensive global database of carbon fluxes is essential for understanding the Earth's carbon cycle. The primary methods of obtaining the spatial distribution of land carbon fluxes include utilizing machine learning models based on in situ measurements, estimating through satellite remote sensing, and simulating ecosystem models. Recently, an innovative machine learning product known as the Global Carbon Flux Dataset (GCFD) has been released. In this study, we assessed the reliability of the GCFD by comparing it with existing data products, including two machine learning products (FLUXCOM and NIES (National Institute for Environmental Studies)), two ecosystem model products (TRENDY and EC-LUE (eddy covariance–light use efficiency model)), and one remote sensing product (Global Land Surface Satellite), on both site and global scales. Our findings indicate that, in terms of average absolute difference, the spatial distribution of the GCFD is most similar to the NIES product, albeit with slightly larger discrepancies compared to the other two types of products. When using site observations as the benchmark, gross primary production (GPP), respiration of ecosystem (RECO), and net ecosystem exchange of machine learning products exhibit higher $R^2$ (ranging from 0.57 to 0.85, 0.53–0.79, and 0.31–0.70, respectively) compared to model products and remote sensing products. Furthermore, we analyzed the spatial and temporal distribution characteristics of carbon fluxes in various regions. The results demonstrate an upward trend in both GPP and RECO over the past two decades, while NEE exhibits an opposite trend. This trend is particularly pronounced in tropical regions, where higher GPP is observed in tropical, subtropical, and oceanic climate zones. Additionally, two remote sensing variables that influence changes in carbon fluxes, i.e., fraction absorbed photosynthetically active radiation and leaf area index, exhibit relatively consistent spatial and temporal characteristics. Overall, our study can provide valuable insights into different types of carbon flux products and contribute to understanding the general features of global carbon fluxes.

**Keywords:** carbon fluxes; gross primary production; terrestrial ecosystem respiration; net ecosystem exchange

## 1. Introduction

Understanding the intricate dynamics of land carbon fluxes is essential in grasping the Earth's carbon cycle and its implications for global climate change [1,2]. There are several types of carbon fluxes, such as gross primary production (GPP), terrestrial ecosystem respiration (RECO), and net ecosystem exchange (NEE). GPP signifies the total assimilation of carbon dioxide by plants through photosynthesis, while RECO represents the release of carbon dioxide by ecosystems through respiration processes [3,4]. On the other hand, NEE refers to the net carbon exchange between ecosystems and the atmosphere, which can be positive (carbon release) or negative (carbon uptake) [5,6]. These crucial carbon fluxes play a significant role in regulating the global carbon balance and are influenced by diverse factors such as climate, vegetation type, and land use changes [7,8].

To enhance our understanding of land carbon fluxes and their intricate interactions, researchers have employed diverse datasets obtained through various methodologies. These include remote sensing observations, ecosystem modeling, and machine learning techniques [9]. In recent years, advancements in remote sensing technology have significantly contributed to our knowledge of global photosynthesis and to theories, simulations, and observations of carbon fluxes. Among the widely used remote sensing products is the MODIS satellite, which provides global GPP products at 8-day intervals, with a spatial resolution of 500 m, based on light use efficiency (LUE) [10]. To address the errors and uncertainties arising from mismatched input data scales and data quality issues, Zhao et al. modified the input meteorological data of MODIS products and other relevant parameters [11]. Additionally, the VPM (vegetation photosynthesis model) remote sensing product utilized an improved light use efficiency model, incorporating remote sensing variables and NCEP (National Centers for Environmental Prediction) reanalysis data. The algorithm related to vegetation indices was also enhanced, thereby addressing the limitations of previous GPP products [12]. Another valuable remote sensing product, the Global Land Surface Satellite product (GLASS), based on remote sensing data, offers high-resolution global data for 12 variables, dating back to 1981. Notably, GLASS data exhibit good spatial continuity and contain no missing values [13].

Many ecosystem models have been developed to describe the land carbon cycle in ecosystems, and these models can also respond to different climatic and atmospheric conditions [14]. Models based on light use efficiency have proven effective in improving estimates of GPP [8,15]. For example, Yuan et al. used an eddy covariance–light use efficiency model to generate high-resolution GPP products, accurately representing both spatial and temporal patterns [16–18]. A process-based model combining atmospheric, radiative, photosynthetic, transport, and energy balance factors was used to estimate global GPP and evapotranspiration (ET) by utilizing FLUXNET 2015 data and MODIS data, achieving the same spatial resolution as the MODIS product [19,20]. Moreover, Li et al. addressed the uncertainty in estimating GPP from solar-induced chlorophyll fluorescence (SIF) by exploring eight different forms of the relationship between SIF and GPP. They obtained a high-resolution GPP global dataset at 0.05° resolution, which exhibited a strong fit to the observed data [21]. Gao et al. developed a remote sensing-based model for predicting RECO driven by multiple MODIS remote sensing data in northern China and the Tibetan Plateau region [22]. Richardson et al. presented three models for predicting RECO and employed statistical methods to assess model and parameter uncertainties, providing valuable references for RECO modeling in terms of data and methods [23]. Ge et al. developed a predictive model for RECO in grassland ecosystems in northern China, integrating factors such as temperature, humidity, and productivity. They used MODIS surface data to simulate RECO, offering a comprehensive understanding of the climatic and environmental influences on RECO in grassland ecosystems [24]. Moreover, Mendes et al. estimated seasonal variations in GPP, RECO, and NEE in a tropical forest in Brazil. Their results revealed that carbon fluxes were influenced by precipitation and vegetation distribution, while NEE remained balanced even during the dry season [25]. Dyukarev et al. employed soil and air temperature, radiation, and leaf area index (LAI) to model and estimate the distribution of

GPP, RECO, and NEE in western Siberia, demonstrating that the studied ecosystem acted as a carbon sink [26]. Fang et al. investigated the relationship between carbon fluxes and climate variables during different phases of the El Niño–Southern Oscillation (ENSO) by utilizing carbon flux outputs from a global ecosystem model [27]. Zhou et al. used the biogeochemical Carnegie Ames Stanford Approach (CASA) model to develop a dataset of GPP, RECO, and NEE for North America [28].

Data-driven machine learning methods are playing an increasingly significant role. This approach typically uses satellite remote sensing data, meteorological data, and other datasets related to land carbon fluxes to construct machine learning models. Numerous research projects have utilized a variety of machine learning models to forecast carbon fluxes. Among these projects, the FLUXCOM project has been widely adopted [29]. The model tree ensemble approach, which relies on remote sensing and meteorological data, represents a significant advancement in global land carbon flux estimation [30]. The resulting dataset can be used to evaluate existing models of land surface processes and the biosphere's condition [31]. Tramontana et al. evaluated the performance of a random forest model for different target variables across various regions [32]. Their study employed 11 machine learning algorithms from four major classes, including tree models, kernel function methods, neural networks, and regression splines. The outputs were compared to the results of existing models [33,34]. Prediction results using a random forest model demonstrated its superiority over the MODIS product in certain areas [35]. Xiao et al. utilized a modified regression tree to predict net ecosystem carbon exchange in the United States [36]. A cubic regression tree model based on MODIS remote sensing data and reanalysis meteorological data was used to predict global SIF and GPP [37]. A tree ensemble model was also employed to generate a national dataset for China [38]. Zeng et al. employed a random forest model to upscale GPP, RECO, and NEE data from the FLUXNET2015 dataset to derive global carbon flux products spanning 1999 to 2019 [39]. Neural network models employing remote sensing SIF, along with other radiometric and meteorological data as predictors, have also been employed to predict global GPP [40]. Furthermore, a support vector regression model can produce GPP and NEE products on both site and spatial scales within Asia [41].

For the aforementioned three types of land carbon flux products, it is necessary to evaluate the carbon fluxes derived from their predictions due to the differing methodologies employed by these products. With the rapid advancement of machine learning technology in carbon flux simulation, a notable example being the recently released Global Carbon Flux Dataset (GCFD) product [42,43], it is crucial to also examine the reliability of products generated using this technology. This study aimed at comparing three distinct types of products, including machine learning products, ecosystem model products, and remote sensing products, to further assess whether the carbon flux products obtained through machine learning algorithms adequately reflect the realistic distribution of carbon fluxes. The objective was to enhance the reliability and applicability of carbon flux products predicted by machine learning algorithms. Furthermore, in order to conduct a comprehensive analysis of global carbon flux characteristics, this study analyzed temporal variations and spatial distributions of global carbon fluxes. Additionally, an attribution analysis of the spatial and temporal variations in carbon fluxes was performed, taking into account significant factors that influence changes in carbon fluxes.

The remaining structure of the paper is organized as follows. Section 2 describes the carbon flux products and evaluation methods. Section 3 gives the comparison results of different products and an in-depth analysis of global carbon flux distribution. Sections 4 and 5 present the discussion and conclusions, respectively.

## 2. Materials and Methods

This study analyzed three types of land carbon fluxes, including GPP, RECO, and NEE. They were chosen according to previous studies and variables provided in the datasets.

### 2.1. Flux Tower Site Data

Observations from three flux tower datasets were utilized for site-level validation in this study. These datasets include FLUXNET2015 [44], FLUXNET-CH4 [45], and Drought-2018 [46], collectively comprising a total of 280 sites worldwide. They provide various variables such as GPP, RECO, and NEE. Initially, daily values were extracted from these datasets, spanning the timeframe from 1999 to 2018. Subsequently, the extracted data were aggregated into monthly averages to ensure their consistency with the time intervals of other products.

### 2.2. Machine Learning Products

Three datasets, including the GCFD (Global Carbon Flux Dataset) [42], the FLUXCOM RS+METEO product [33,34], and the NIES (National Institute for Environmental Studies) product by Zeng et al. [39], were utilized for evaluation in this study. The GCFD is a global land carbon flux dataset with a resolution of 1 km, generated using deep learning techniques trained with in situ measurements from 280 stations, which are the same as the flux tower sites in Section 2.1. It encompasses three carbon flux variables, namely gross primary production (GPP), terrestrial ecosystem respiration (RECO), and net ecosystem exchange (NEE), recorded at 10-day intervals from January 1999 to June 2020. FLUXCOM provides monthly data on the three carbon fluxes, with a spatial resolution of 0.5°, covering the period from 2001 to 2010. Zeng et al. provided NIES data on the three carbon fluxes at a 10-day resolution and a spatial resolution of 0.1°, spanning from 1999 to 2019. Compared to FLUXCOM and NIES, the GCFD is derived using deep learning with better performance than traditional machine learning; the GCFD demonstrated more spatial details with higher resolution.

### 2.3. Ecosystem Model Products

Two model products, namely the TRENDY model product [47] and the EC-LUE (eddy covariance–light use efficiency) model product by Yuan et al. [16,48], were utilized as the representative ecosystem model products in this study. GPP data from 1999 to 2018 from these two models were employed. The TRENDY model offers monthly data with a spatial resolution of 1°, while Yuan et al.'s model provides GPP data every 8 days at a spatial resolution of 0.05°.

### 2.4. Remote Sensing Product

The Global Land Surface Satellite (GLASS) product [13,49,50] was utilized as the representative remote sensing carbon flux product. This comprehensive product incorporates 14 different land variables, encompassing the prevailing LUE models presently accessible. The GPP variable obtained from GLASS offers a spatial resolution of 0.05° and a temporal resolution of 8 days. For the purposes of this study, data spanning from 1999 to 2018 were employed for evaluation.

### 2.5. Methods for Data Comparison

Detailed information about the carbon flux datasets has been summarized in Table 1. Here, the GCFD was compared to land carbon flux products provided by Zeng et al., FLUXCOM, TRENDY, Yuan et al., and GLASS. No additional quality check was performed, and all data are used for the comparison. It should be noted that the resolution of GCFD (1 km) is closer to the footprint area of eddy covariance data, which makes it more comparable to in situ data than the other products with coarser resolution. When different datasets are compared, the product with higher spatiotemporal resolution is processed according to the product with lower resolution using the averaging method. Since the FLUXCOM data are available on a monthly basis, all datasets were processed to a monthly scale for accurate comparison. Since only GPP is available in the carbon flux data from TRENDY, Yuan et al., and GLASS, the focus of the comparison was solely on the differences in GPP predictions between the GCFD and the aforementioned datasets. Also, in addition to the global scale, the analysis was also conducted in seven climate zones, including the tropical, subtropical,

continental, Mediterranean, oceanic, dry, and polar climates, which were divided according to the Koppen climate classification [51].

**Table 1.** Detailed information on global carbon flux datasets.

| Dataset Name | Spatial Resolution | Temporal Resolution and Coverage | Estimating Method |
|:---:|:---:|:---:|:---:|
| GCFD | 1 km | Every 10 days from January 1999 to June 2020 | |
| FLUXCOM | 0.5° | Monthly, from 2001 to 2010 | Machine learning |
| NIES | 0.1° | Every 10 days from 1999 to 2019 | |
| TRENDY | 1° | Monthly, from 1999 to 2018 | Ecosystem model |
| EC-LUE | 0.05° | Every 8 days from 1999 to 2018 | |
| GLASS | 0.05° | Every 8 days from 1999 to 2018 | Remote sensing |

*2.6. Product Evaluation Methods*

In station-level validation, this study employed the observations from each station to compare them with the predictions generated by the products at the corresponding station locations. The coefficient of determination ($R^2$) [52], root mean square error ($RMSE$) [52], and bias [52] were used to evaluate the performance of products as:

$$R^2 = 1 - \frac{\sum_{i=1}^{n}(y_i - \hat{y}_i)^2}{\sum_{i=1}^{n}(y_i - y)^2},$$ (1)

$$RMSE = \sqrt{\frac{\sum_{i=1}^{n}(y_i - \hat{y}_i)^2}{n}},$$ (2)

$$bias = \frac{\sum_{i=1}^{n}(\hat{y}_i - y_i)}{n}$$ (3)

where $y_i$ and $\hat{y}_i$ are the observed and predicted values, respectively, and $y$ is the averaged value of observations.

*2.7. Attribution Analysis*

To identify significant factors among the variables associated with carbon fluxes, this study employed two methods for attribution analysis; the corresponding results are presented in a study by Shangguan et al. [42]. The first method involved calculating all correlation coefficients between the predictors and the three target variables to identify the variables with high associations. The second method utilized a random forest model to determine the feature importance of the predictors and rank them accordingly. The first and second method can identify the linear and nonlinear relationships for attribution analysis, respectively. Following attribution analysis, four important variables were selected for further analysis in Section 3.4, including FAPAR (fraction of absorbed photosynthetically active radiation), LAI, air temperature (TA), and latent heat flux (LE). FAPAR and LAI were obtained from the Copernicus Global Land Service (CGLS) [53], while air temperature and latent heat flux were obtained from ERA5-Land [54] with a spatial resolution of 0.1 degree and temporal resolution of 3 h.

**3. Results**

*3.1. Comparisons of Products*

Comparisons were conducted using data from each flux tower site and the corresponding grid data to obtain the results presented in Figure 1. The findings indicate that the GCFD demonstrated the most outstanding performance, whereas the predictions of the other datasets exhibited notable underestimations or overestimations. Evaluating the $R^2$ and $RMSE$ values, FLUXCOM and Zeng et al. showcased very similar performance, while the GCFD exhibited higher $R^2$ values and lower $RMSE$ values for all three predicted fluxes.

Notably, when compared to FLUXCOM, the GCFD exhibited a remarkable enhancement in $R^2$, with improvements of 49%, 49%, and 126% for GPP, RECO, and NEE, respectively. Among all the products, GCFD predictions exhibited the lowest biases, with GPP, RECO, and NEE showing prediction biases of −0.044, 0.018, and 0.053, respectively. Conversely, FLUXCOM displayed the most pronounced bias.

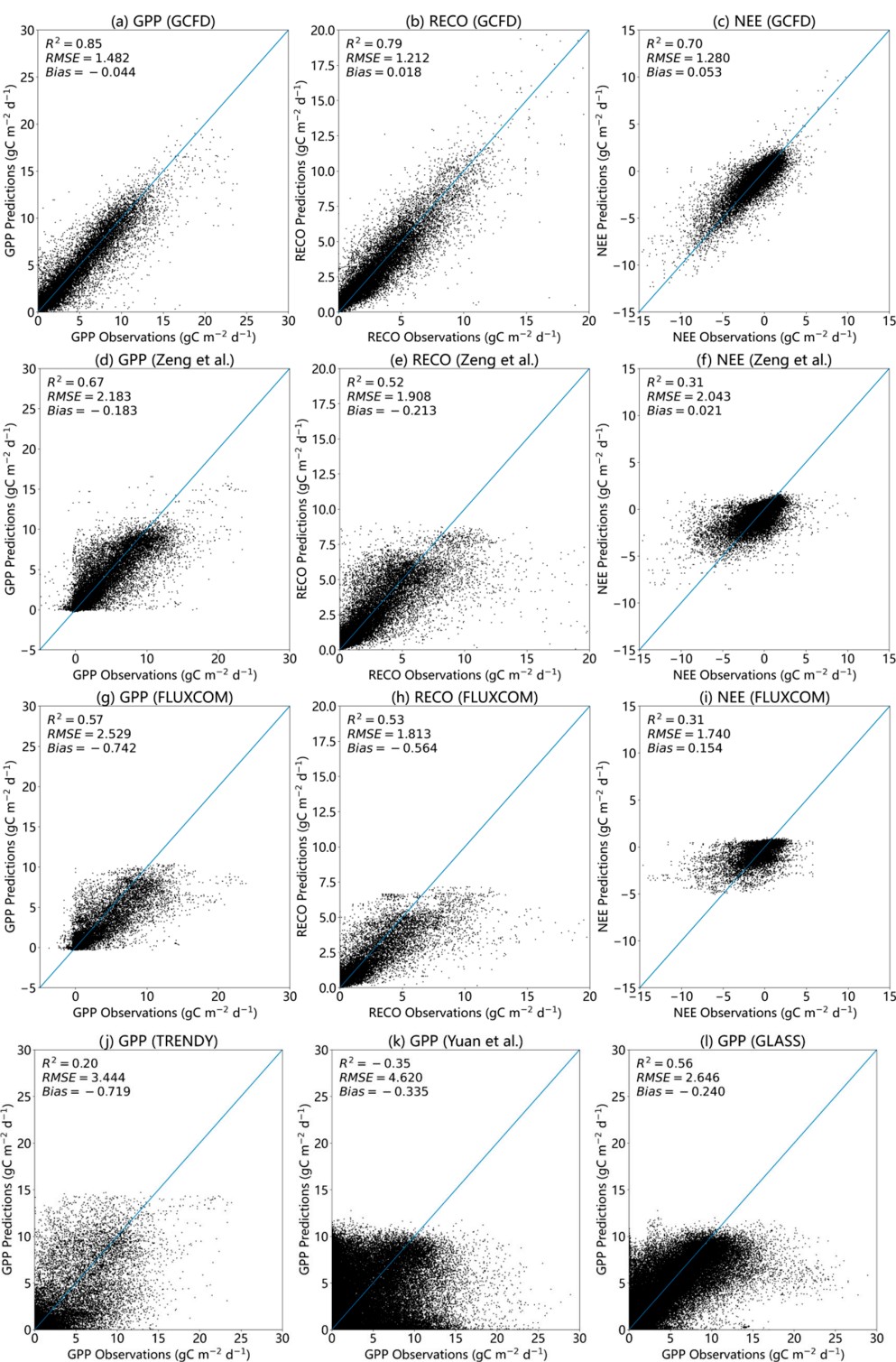

**Figure 1.** Scatter plots between observed and predicted values for GPP, RECO, and NEE from GCFD (**a**–**c**), Zeng et al. (**d**–**f**), and FLUXCOM (**g**–**i**), and GPP from TRENDY (**j**), Yuan et al. (**k**), and GLASS (**l**).

The discrepancy between the predicted carbon flux values of the ecosystem model and the in situ observations is more pronounced in comparison to the machine learning products. This disparity could arise from a mismatch in the scales of the model and the site, possibly due to the low spatial resolution of the model data. Additionally, it could be attributed to the uncertainties generated by the model during the parameterization and simplification of the ecosystem structure.

Compared to ecosystem model products, the remote sensing product produced GPP data that are closer to site data and exhibited significantly smaller bias. The validation accuracy stands at 0.56, though it remains lower than the site validation accuracy of machine learning. Remote sensing products, derived primarily from satellite-observed data, offer the advantage of reducing complex calculation processes and minimizing data generation uncertainties. However, satellite observations are subject to environmental and equipment conditions, leading to greater uncertainty compared to machine learning methods, which rely solely on data. These factors suggest that the accuracy of carbon flux products estimated through remote sensing methods falls between that of ecosystem model products and machine learning products.

A further comparison was conducted by subtracting the 2010 annual averages of the three carbon fluxes predicted by the GCFD from the predicted values of other products. This process generated a distribution of differences, which is illustrated in Figure 2. Among the three carbon fluxes, NEE exhibited the smallest disparity across the data products due to its relatively narrow range of values. The GCFD only exhibited a slight overestimate or underestimate in tropical regions when compared to the two machine learning products. The distinction between GCFD and FLUXCOM predictions for RECO was more prominent in the Amazon region of South America. On the other hand, the gap between the GCFD and FLUXCOM was slightly larger than that between the GCFD and Zeng et al. In the case of GPP, the GCFD tended to slightly overestimate FLUXCOM in various regions, such as Siberia, North America, and South America, where the difference between the GCFD and Zeng et al. was smaller. GCFD predictions in tropical Africa were slightly lower than the two machine learning datasets, but the underestimation was relatively minor. These outcomes might be attributed to the similarity in the covariates used in the GCFD modeling process compared to those employed by Zeng et al., resulting in closer predictions.

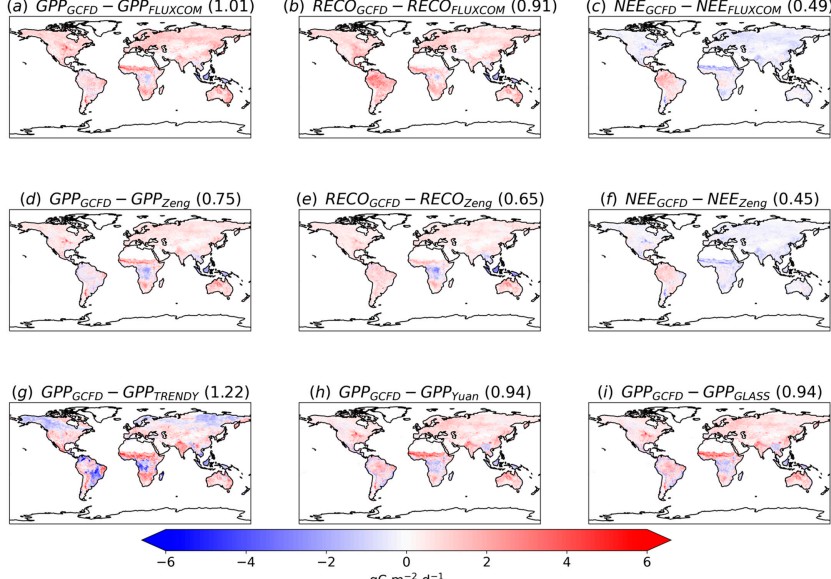

**Figure 2.** Distribution of the differences between the three carbon fluxes predicted by the GCFD and the predictions of FLUXCOM (**a**–**c**) and Zeng et al. (**d**–**f**); differences between GPP predicted by GCFD and TRENDY (**g**), Yuan et al. (**h**), and GLASS (**i**), with the data in the plots averaged over the year 2010. Values in brackets are average absolute differences between products.

The GPP values predicted by the GCFD exhibited varying degrees of over- or underestimation in certain regions when compared to ecosystem model data. In tropical Africa, the GPP values predicted by GCFD were slightly lower than those obtained by both models. Similarly, in southern Asia and certain coastal areas of South America, the GCFD predictions were also lower compared to the model predictions. Conversely, in the remaining regions across the globe, the GCFD predictions for GPP tended to be higher than the model predictions to varying extents. The most substantial differences were primarily observed in areas adjacent to the southern Sahara, the northern part of the Indochinese Peninsula, the central United States, and some regions of south-central South America. Overall, the disparity between GCFD and ecosystem models was more pronounced than that between machine learning products.

The GCFD product exhibited significantly higher values in the Amazon region compared to GLASS. The range of high values obtained by GCFD was greater than that of GLASS in the high-value region of southeastern China. In the eastern USA and western Europe, the GCFD exhibited similar characteristics. However, in tropical Africa and Southeast Asia, GLASS showed more prominently high values. Overall, the GCFD product effectively captured the accurate distribution of GPP in most regions.

Based on the global absolute difference, it is evident that the smallest disparity was found between the GCFD and Zeng et al. In comparison to machine learning products, the absolute values between the GCFD and the remaining datasets were relatively higher. Consequently, it can be inferred that the GCFD exhibits the highest similarity to Zeng et al. among the three categories of carbon flux products.

*3.2. Characteristics of Temporal Variation in Carbon Fluxes*

Carbon fluxes have a tendency to fluctuate over time due to various factors such as climate change, vegetation conditions, and meteorological factors. These fluxes exhibit different patterns of change in different regions across the world. Figure 3 illustrates the trends in carbon fluxes based on the GCFD, showing the average annual change in each carbon flux variable from 1999 to 2020. The findings indicate that, among the three carbon flux variables, GPP and RECO demonstrated consistent trends over time, while NEE showed an opposite trend to GPP and RECO. The analysis primarily focused on the trend of GPP. Regarding regional variations, significant changes in carbon fluxes over time were observed in central Africa and near the tropics of South America, where GPP exhibited a clear upward trend. The northern regions of Asia and Europe also displayed a slight upward trend. Conversely, the central region of the USA demonstrated the most noticeable downward trend in GPP. Additionally, some areas on the edge of the Sahara desert and in certain parts of the Far East exhibited downward trends as well. A comparative analysis revealed that the extent of rising GPP in the tropics surpassed the areas where declining GPP was observed, indicating a more pronounced global trend of increasing GPP over time.

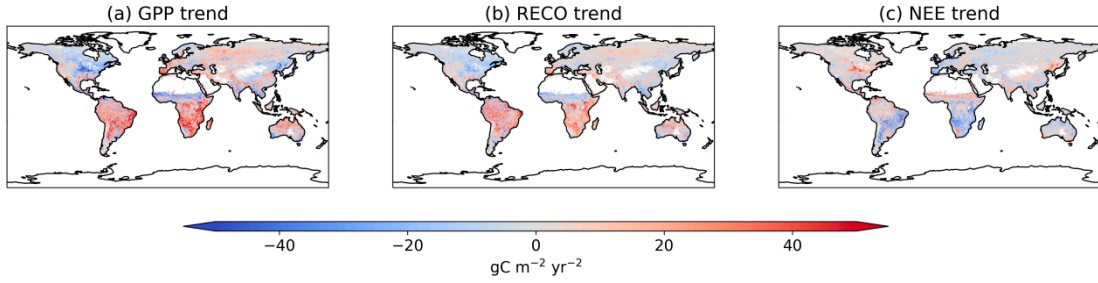

**Figure 3.** Spatial distribution of annual average carbon flux trends from 1999 to 2020 in the GCFD dataset, with positive values indicating increases and negative values indicating decreases.

Next, this study calculated the mean carbon fluxes over time on both the global scale and within seven climate regions, as depicted in Figure 4. Regarding the global time series, the annual sum of GPP and RECO exhibited a slight increase over time, while the value of

NEE declined. Specifically, the global mean GPP rose from 1158.1 g C m$^{-2}$ yr$^{-1}$ in 1999 to 1217.8 g C m$^{-2}$ yr$^{-1}$ in 2019, and RECO increased from 995.0 g C m$^{-2}$ yr$^{-1}$ in 1999 to 1025.3 g C m$^{-2}$ yr$^{-1}$ in 2019. Conversely, NEE decreased from −215.6 g C m$^{-2}$ yr$^{-1}$ in 1999 to −244.6 g C m$^{-2}$ yr$^{-1}$ in 2019. It is plausible that the upward trend in global GPP is attributable to the recent increase in greenhouse gases and rising temperatures. Analyzing the time series of individual climate zones, a more pronounced trend of increasing GPP values can be observed in the tropical, Mediterranean, subtropical, continental, and polar climate zones. However, in the dry and oceanic climate zones, the regional averages of carbon fluxes fluctuated relatively widely over time. Overall, the general trend in GPP indicates a predominant increase over time, both globally and within individual regions.

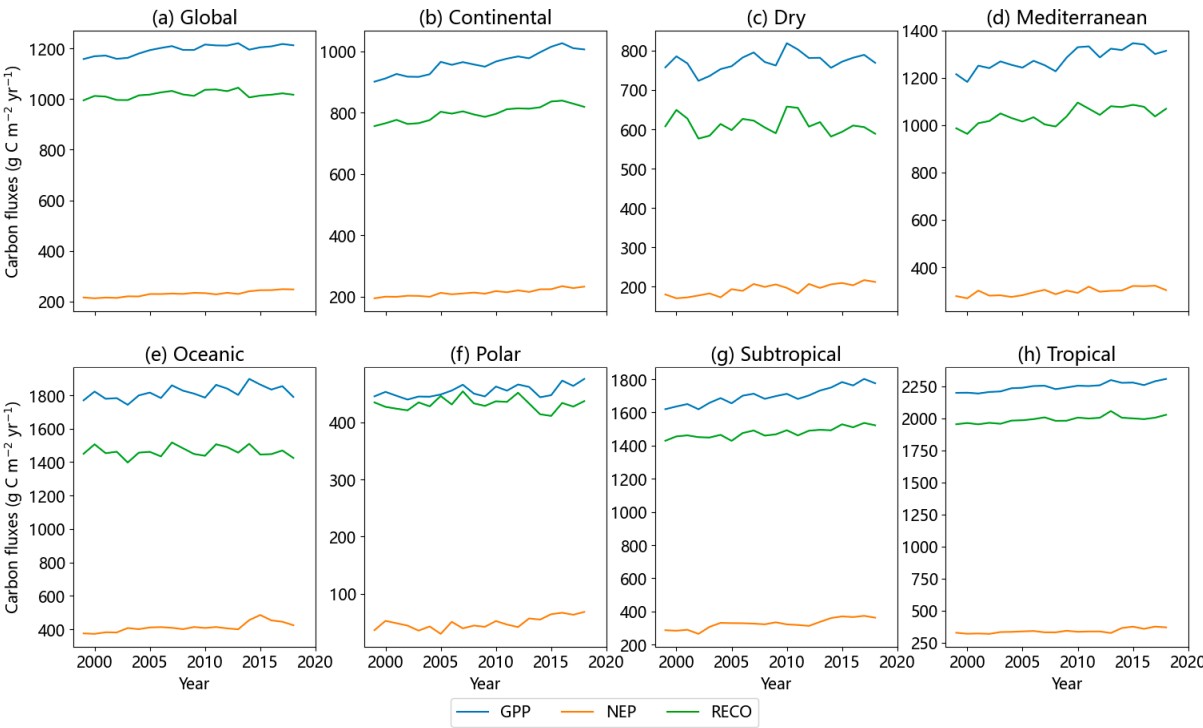

**Figure 4.** Variation in annual sums of GPP, RECO, and NEP (net ecosystem production), calculated from the GCFD dataset, global and for seven climate zones over the period from 1999 to 2019. We provided NEP instead of NEE here for the convenience of plotting, which is the opposite value of NEE.

The global distribution patterns of carbon fluxes for the four seasons are presented in Figure 5. It is evident that regardless of the seasons, GPP exhibits its highest values near the equator and decreases with increasing latitude. In regions such as central Africa, the Amazon area, and the South Asian tropics, GPP values remain consistently high throughout the year. In certain temperate regions, there is a noticeable seasonality in GPP variations. For instance, in northern Asia and Europe, the eastern United States, and southeastern China, GPP values reach their peak during summer and hit their lowest point in winter. This disparity may be attributed to the contrasting vegetation types found in tropical and temperate areas. Vegetation in the tropics tends to remain evergreen due to favorable climatic conditions, while the climate in temperate regions undergoes substantial changes across seasons, resulting in phenomena like defoliation and a significant decline in productivity during the winter. Regarding the seasonal fluctuations of RECO, the spatial distribution maps for the four seasons demonstrate a relatively similar pattern to that of GPP. Likewise, the spatial distribution maps for NEE throughout the four seasons indicate that central Africa and the southern Asian tropics consistently function as carbon sinks throughout the year. Conversely, areas such as the eastern United States, Europe,

and East Asia exhibit higher carbon absorption during summer, while certain regions in South America and southern Africa act as carbon sinks during winter and spring. From Figures 3 and 5, it is clear that in some countries with fast-paced development, such as China and India, carbon fluxes showed obvious variations over seasons and trends over the years, driven by factors such as aerosol, precipitation, and temperature [55–57].

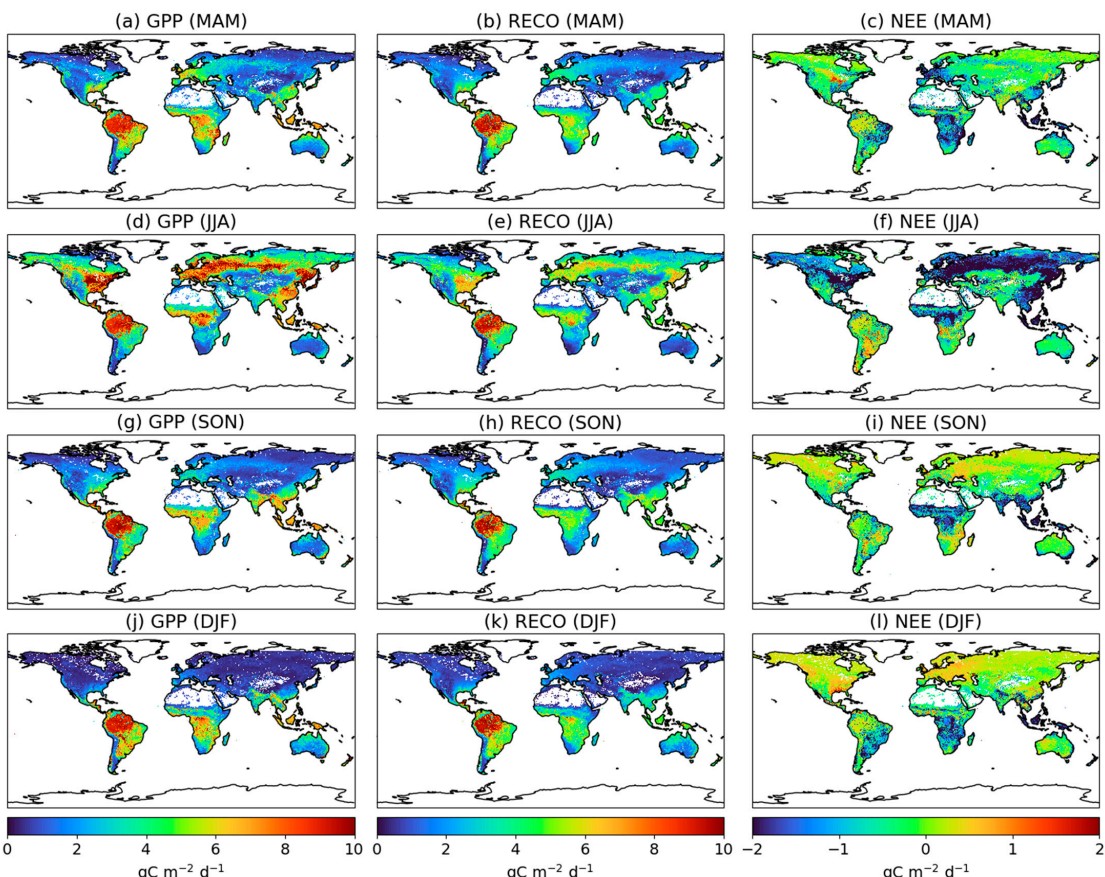

**Figure 5.** Seasonal average distribution of three carbon fluxes predicted by GCFD. MAM: March–May; JJA: June–August; SON: September–November; DJF: December–February.

### 3.3. Characteristics of Spatial Variation in Carbon Fluxes

The spatial distribution pattern of carbon fluxes can be influenced by the environmental conditions of different regions, resulting in variations in values. In this study, the global distribution of carbon fluxes and their distribution within seven distinct climate zones were calculated, as depicted in Figure 6. This approach enables a more quantitative comparison of the spatial distribution characteristics of carbon fluxes. On a global scale, GPP (gross primary production) and RECO (ecosystem respiration) values are primarily concentrated within the range of 0–8 g C m$^{-2}$ d$^{-1}$, while NEE (net ecosystem exchange) values are mainly concentrated within the range of −2–2 g C m$^{-2}$ d$^{-1}$. The median values for GPP, RECO, and NEE are 2.6, 2.2, and −0.5 g C m$^{-2}$ d$^{-1}$, respectively. GPP exhibits the largest range of variation, followed by RECO, and NEE shows the smallest range of variation. The results for different climatic zones indicate that the tropics exhibit the highest GPP among all climatic zones, with a median GPP of 5.9 g C m$^{-2}$ d$^{-1}$. This finding aligns with the tropical climatic environment, characterized by a hot and humid climate, ample sunshine, and abundant vegetation, which favor vegetation productivity. Additionally, subtropical regions with favorable climatic and vegetation conditions as well as oceanic climatic zones with mild climates also display higher GPP values, with a median of 4.6 g C m$^{-2}$ d$^{-1}$. Conversely, polar regions, characterized by cold climates and sparse vegetation, exhibit the lowest GPP values, with a median of only 1.0 g C m$^{-2}$ d$^{-1}$. Furthermore, in dry climate

zones, where moisture conditions are insufficient, and in certain areas such as deserts with sparse vegetation, GPP values are relatively low, with a median of 1.7 g C m$^{-2}$ d$^{-1}$. Overall, the distribution of high and low carbon flux values is closely linked to climatic conditions, primarily temperature, moisture, and vegetation type.

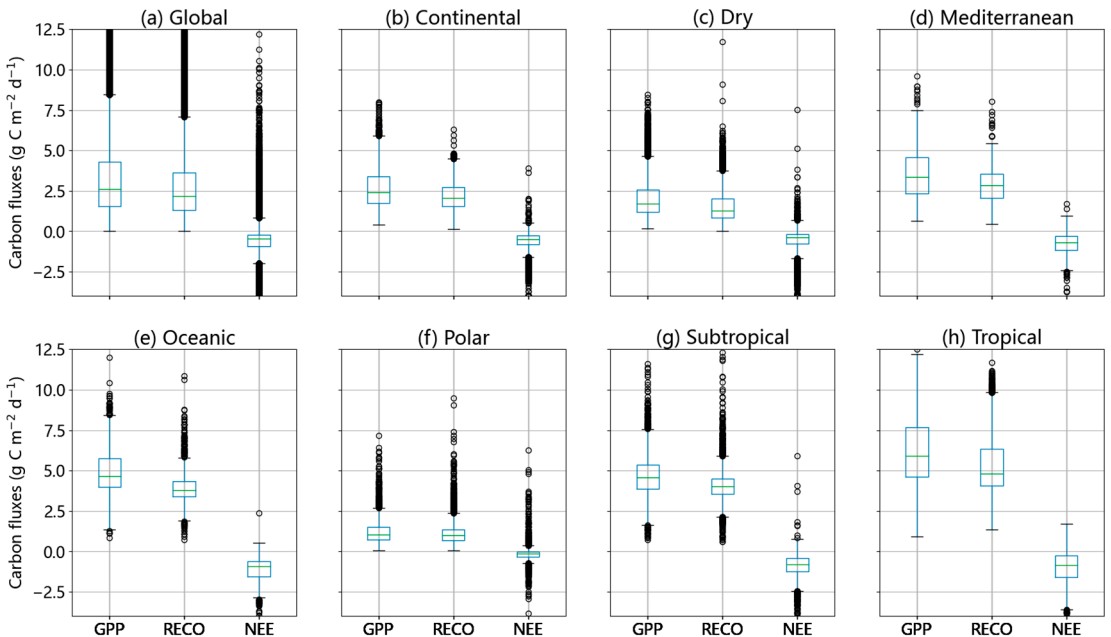

**Figure 6.** Box plot of the distribution of carbon fluxes globally and within each climate zone, with data derived from the average spatial distribution of carbon fluxes from 1999 to June 2020.

### 3.4. Attribution Analysis of the Distribution of Carbon Fluxes

It is crucial to analyze the correlation between the spatial and temporal variability of carbon fluxes and key variables to facilitate the study and prediction of carbon fluxes. For this purpose, we have identified the most significant factors that impact the variability of carbon fluxes, aiming to investigate their relationship with the spatial and temporal variability patterns of carbon fluxes. In this section, we have chosen FAPAR (fraction of absorbed photosynthetically active radiation), LAI, air temperature (TA), and latent heat flux (LE) for attribution analysis.

Figures 7 and 8 calculate the regional averages of the four aforementioned predictors on a global scale and within seven climate zones over the years. In terms of global trends, both remote sensing variables, FAPAR and LAI, exhibit an upward trend over time, with a more notable increase especially after 2010. The global mean of FAPAR increases from 0.30 in 1999 to 0.34 in 2019, while LAI increases from 1.11 in 1999 to 1.24 in 2019. This upward trend aligns relatively consistently with the rising trend in global carbon fluxes. The results for different climate zones indicate that FAPAR and LAI have higher values in tropical and oceanic climate zones compared to other regions. Furthermore, both variables display varying magnitudes of increasing trends within each climate zone, with relatively larger increases in dry, subtropical, continental, and polar climate zones. These findings suggest that the rising FAPAR and LAI are important factors contributing to changes in carbon flux. The time series of temperature, globally and within each climate zone, also exhibit a certain upward trend, indicating a relationship between carbon flux changes and temperature increase. Among the climate zones, the greatest increase in temperature occurs in tropical and subtropical regions. Latent heat fluxes show a decreasing trend over time in the Mediterranean, continental, and polar climatic zones, while in other regions, they mainly demonstrate relatively large fluctuations. Overall, remote sensing variables and temperature changes over time are the primary factors influencing the temporal characteristics of carbon fluxes.

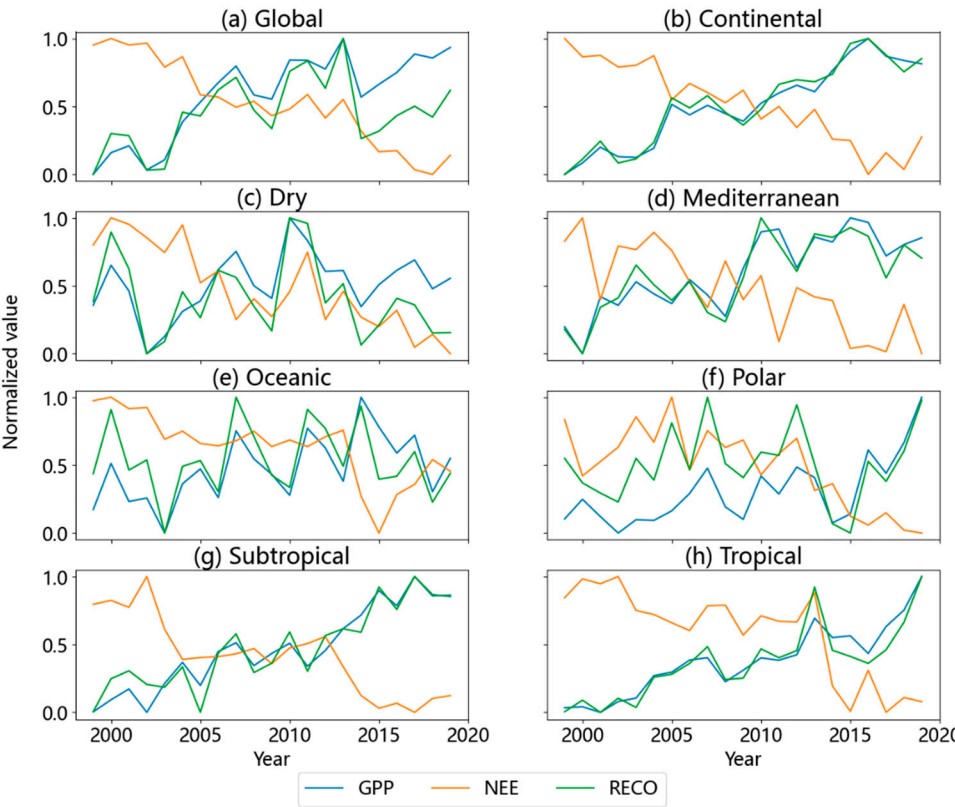

**Figure 7.** Time series of the global average and the regional average within each climate zone for the three carbon fluxes. The variables are normalized for comparison.

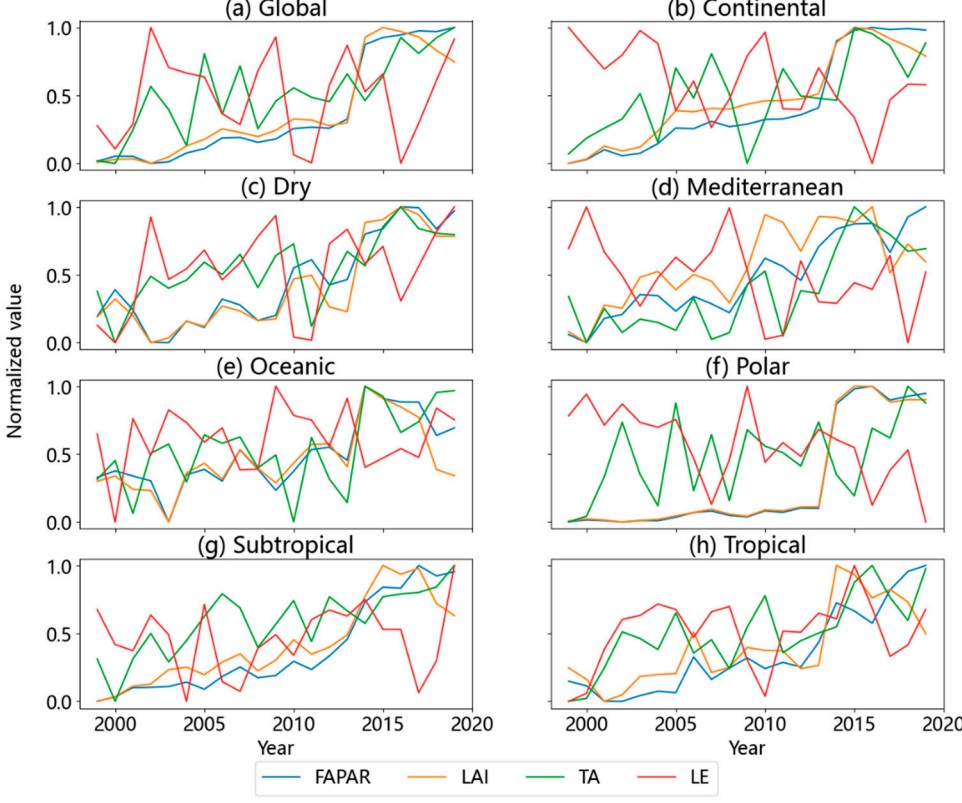

**Figure 8.** Time series of the global average and the regional average within each climate zone for FAPAR, LAI, air temperature, and latent heat fluxes. The variables are normalized for comparison.

The latitudinal average distribution of the three carbon fluxes and four predictors were calculated using the spatial distribution maps, which were obtained by averaging across all time points as depicted in Figure 9. Figure 10 shows the spatial distribution of three carbon fluxes and four predictors. The findings confirm the previous analysis, indicating that GPP and RECO exhibit the highest values at lower latitudes, while carbon sinks are primarily located in the southern hemisphere and the tropics. Specifically, GPP reaches its maximum value of 7.13 g C m$^{-2}$ d$^{-1}$ at 1.5° S, whereas RECO peaks at 6.30 g C m$^{-2}$ d$^{-1}$ at 5.0° S. Conversely, NEE exhibits the lowest value of $-1.27$ g C m$^{-2}$ d$^{-1}$ at 41.5° S. The distribution of the two remote sensing variables reveals that the carbon fluxes attain their peaks in regions that correspond to the peaks of FAPAR and LAI, with both FAPAR and LAI reaching their maximum values at 1° N. Overall, the latitudinal distribution of carbon fluxes aligns well with the remote sensing variables. Furthermore, the latitudinal mean distribution of latent heat fluxes displays a symmetrical relationship with GPP and RECO, with the minimum values of latent heat fluxes coinciding with the maximum values of GPP and RECO. Lastly, the latitudinal averaging of temperature demonstrates the highest values at lower latitudes, gradually decreasing with increasing latitude.

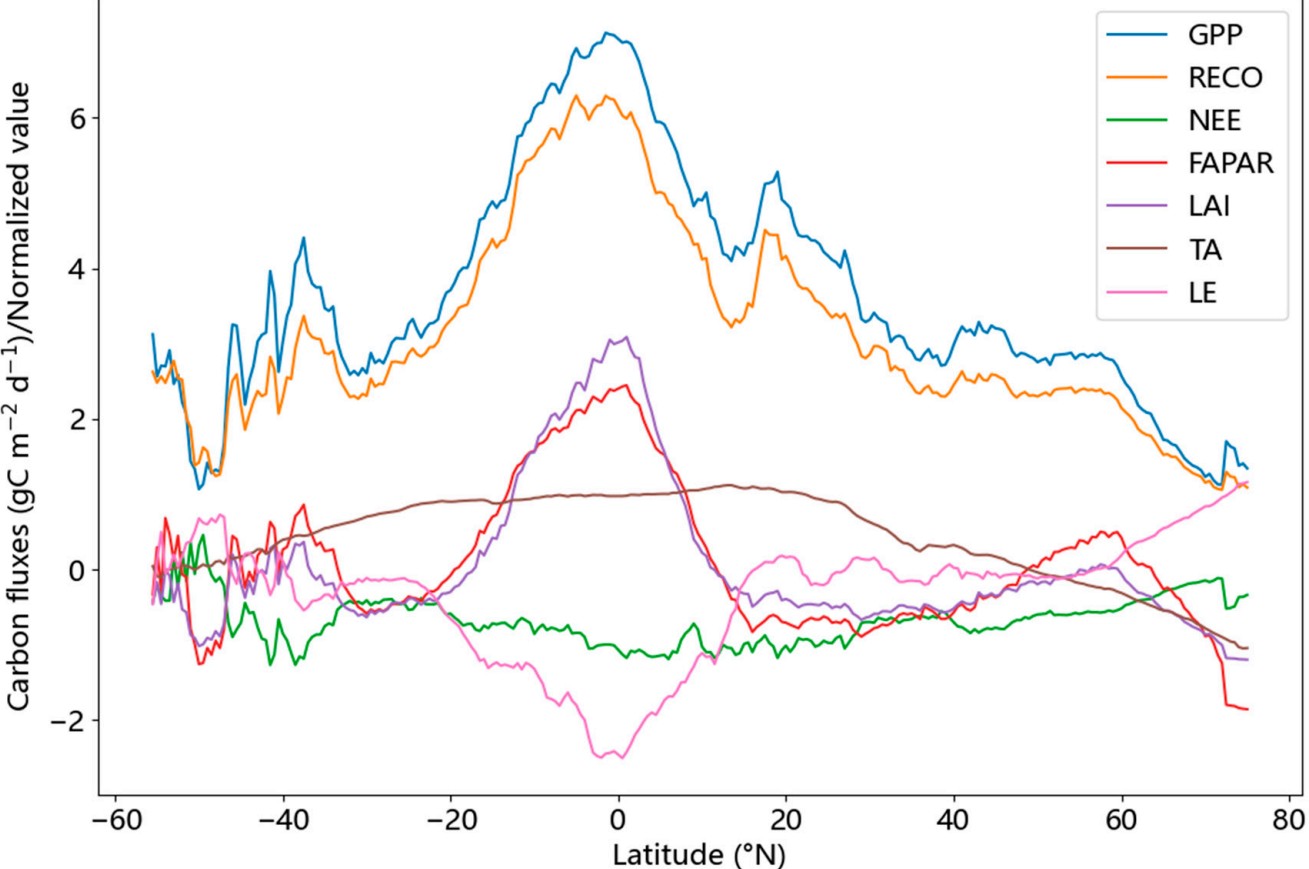

**Figure 9.** Latitudinal mean distribution of carbon fluxes and FAPAR, LAI, air temperature (TA), and latent heat flux (LE), with data derived from the spatially averaged distribution of each variable from 1999 to June 2020, where GPP, RECO, and NEE are in g C m$^{-2}$ d$^{-1}$ and the remaining four predictors have been normalized (subtracted by the average and then divided by the standard deviation).

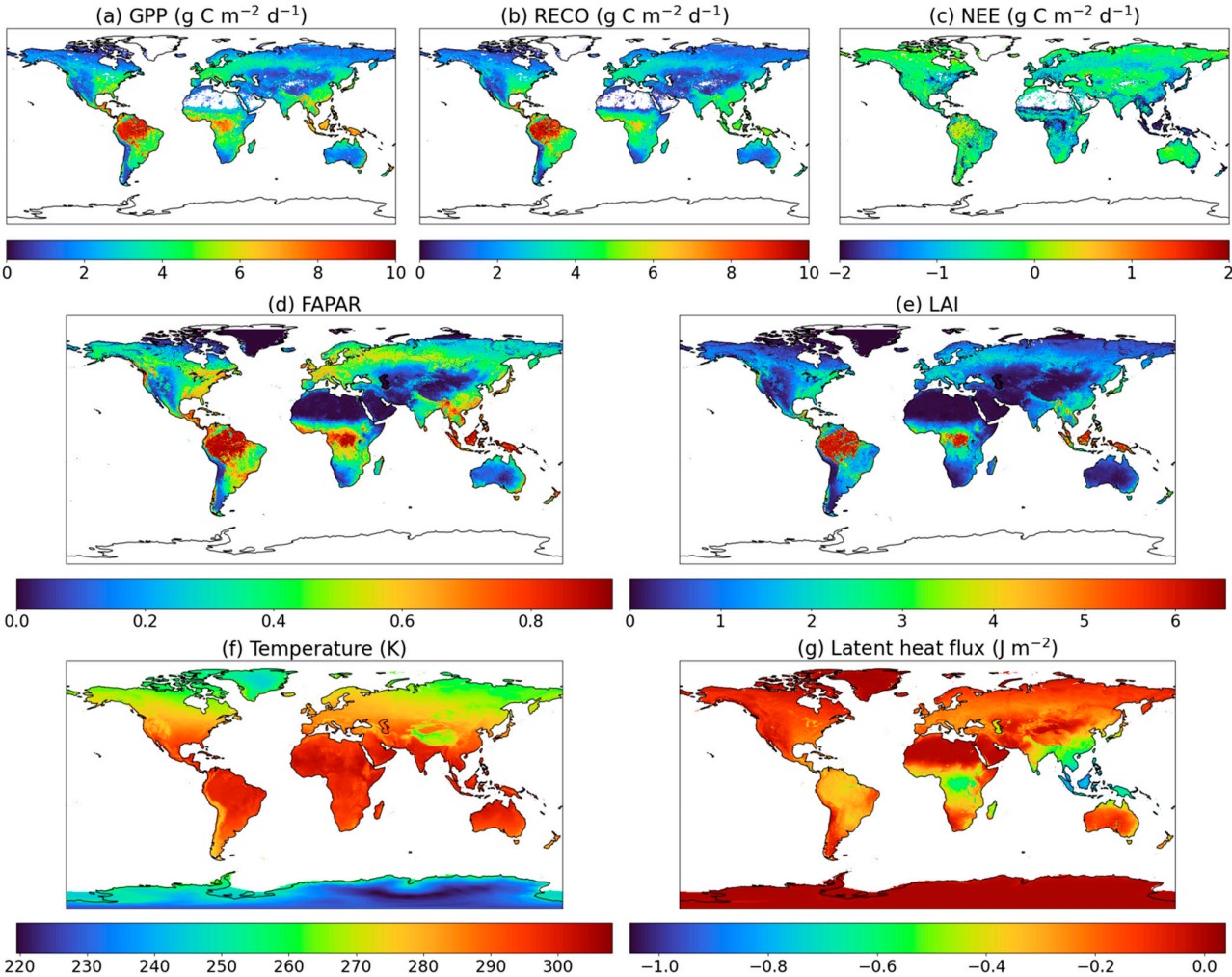

**Figure 10.** Spatial distribution of carbon fluxes (**a**–**c**) and FAPAR (**d**), LAI (**e**), air temperature (TA) (**f**), and latent heat flux (LE) (**g**), with data derived from the spatially averaged distribution of each variable from 1999 to June 2020.

## 4. Discussion

### 4.1. Time Series of Different Products

Figure 11 compares the annual changes in carbon fluxes across different products. Among these products, the carbon fluxes obtained from GCFD predictions are relatively large, second only to TRENDY predictions, and the trends align more closely with those derived from Zeng et al. The GPP and RECO values from the GCFD exhibit an upward trend, accompanied by a decrease in NEE. In the GPP data from TRENDY products, there is also a noticeable upward trend in their year-to-year variation. This result indicates that the interannual variation in carbon fluxes predicted by the GCFD is more pronounced compared to previous studies such as FLUXCOM and GLASS. Though the trends were similar among the GCFD, Zeng et al. and TRENDY, their interannual variations are not exactly the same. For example, the GCFD demonstrated an abrupt decrease in NEE around 2014, while the other two products did not. This abrupt decrease may have been due to the abrupt increase in global LAI and FPAR around 2014, as shown in Figure 8, when vegetation greening led to a carbon sink.

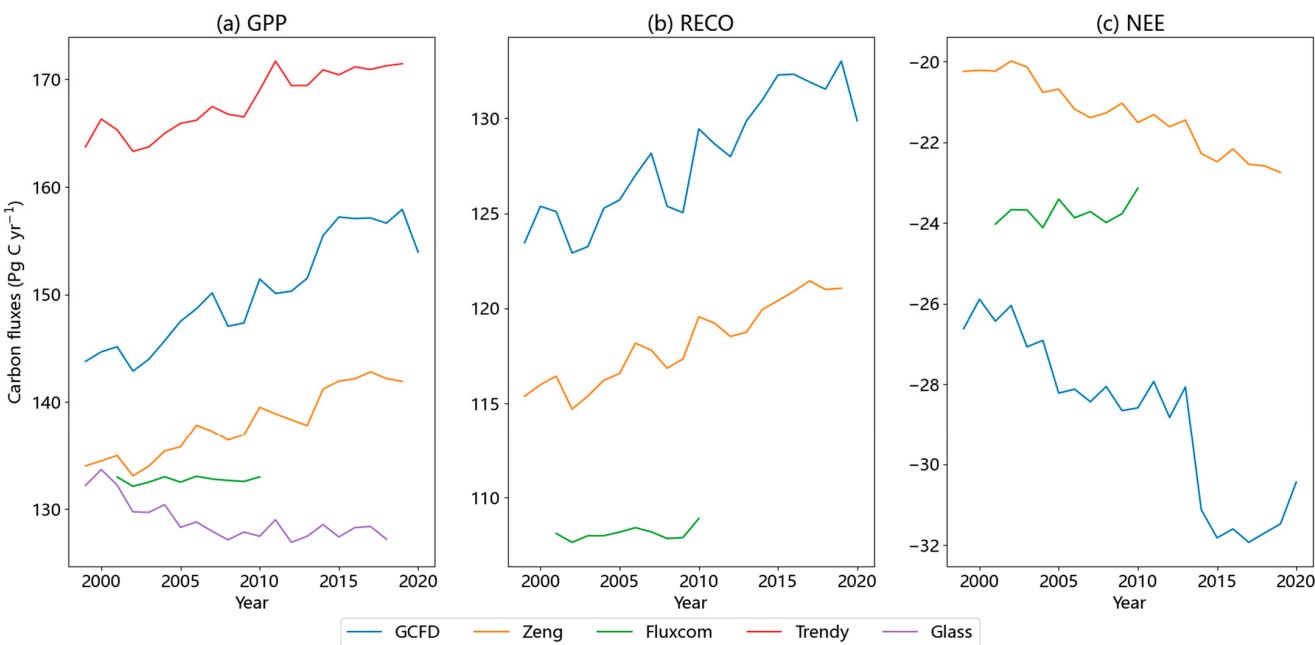

**Figure 11.** Interannual graphs of global carbon fluxes for each product: (**a**) GPP, (**b**) RECO, and (**c**) NEE.

### 4.2. The Depth of Carbon Flux Analysis

There is scope for further enhancement in analyzing the spatial and temporal variability of carbon fluxes in the future. Firstly, focusing solely on the distribution of carbon fluxes in climatic zones overlooks the valuable insights that can be gained from sub-regional approaches, which take into account different ecosystem types. Therefore, it is necessary to consider alternative perspectives, including various ecosystem types, when examining carbon flux distribution. Secondly, the attribution analysis conducted in this study only considered a limited set of predictors, disregarding numerous factors that have an impact on the variability of carbon fluxes. To address this limitation, future studies should aim to conduct more comprehensive attribution analysis by incorporating a wider range of influential factors. This will provide a more comprehensive understanding of the drivers of carbon flux variability.

### 4.3. Representativeness of Sites and Regions

This study utilized carbon flux data from a total of 280 flux tower sites worldwide to validate the data. These sites cover all the seven climate zones and various land cover types in all continents except Antarctica (Table 2). However, it should be noted that the distribution of these sites is not uniform across regions. The number and density of sites are much higher in North America and Europe than in other parts of the world. As a result, site-level validation results are expected to be more accurate and representative within these respective continents. In terms of climate zones, the continental climate zone has the highest number of sites, while the remaining climate zones consist of a similar number of sites ranging from 19 to 33. Concerning land cover types, the majority of sites are located in grassland, savannah, and forests, while only a few sites are located in unvegetated land and cropland. Therefore, it is necessary to gather additional in situ carbon flux data by establishing more flux tower sites in regions where site density is sparse in the future.

**Table 2.** Number of sites in different continents, climate zones, and land cover types.

| Continent | Number | Climate Zone | Number | Land Cover Type | Number |
|---|---|---|---|---|---|
| Asia | 18 | continental | 131 | forest | 65 |
| Africa | 8 | dry | 29 | grassland | 95 |
| North America | 113 | Mediterranean | 33 | shrubland | 20 |
| South America | 8 | oceanic | 21 | cropland | 15 |
| Europe | 109 | polar | 23 | savannah | 74 |
| Oceania | 24 | subtropical | 31 | unvegetated | 11 |
| | | tropical | 19 | | |

*4.4. Limitations of Data Sets*

This study employed three machine learning datasets, two ecosystem model datasets, and one remote sensing dataset to compare site-level and global-level carbon fluxes. However, there are some limitations in this study that affect the assessment of carbon flux product reliability. Firstly, the comparison of carbon flux datasets is inadequate due to the inclusion of only a limited number of products. Numerous datasets on carbon fluxes are available, and their omission restricts the comprehensiveness of the comparison. Secondly, the selected carbon flux products generally have low spatial resolution, which prevents a detailed analysis of the predicted spatial distribution of each dataset in a local area. Consequently, the evaluation of spatial distribution can only be conducted on a larger scale. To improve the reliability assessment of the GCFD product at the regional level, it is necessary to incorporate higher-resolution carbon flux products. Thirdly, the sites used for validation are limited in number and exhibit uneven spatial coverage, and there is a scale mismatch between the site-level observations and the grid-level estimations of carbon fluxes. These factors introduce uncertainties in the validation results of various carbon flux products. Fourthly, despite their ability to represent the actual distribution of carbon fluxes, remote sensing products have their own limitations. Satellite observation can be affected by atmospheric conditions, while factors such as instruments and data processing algorithms can affect remote sensing data, thereby compromising their accuracy and reliability. Therefore, future studies on carbon fluxes require high-quality data to overcome these limitations.

**5. Conclusions**

In order to validate the accuracy of carbon flux predictions from the GCFD generated by a machine learning model and evaluate their capacity to accurately depict the actual spatial distribution of carbon fluxes, this study conducted a comparative analysis of three types of carbon flux products: remote sensing, ecosystem model, and machine learning products. The comparison was conducted on both the site scale and the global scale. Furthermore, this study examined the overall patterns of variation in carbon fluxes by analyzing their time series and spatial distribution. Additionally, an attribution analysis was performed, taking into account significant factors affecting carbon fluxes.

Our conclusions are as follows. First, on the site scale, among the products, the GCFD product shows the closest agreement with site observations in terms of predicted carbon fluxes. Second, the spatial distribution patterns of carbon fluxes are similar across all three product types, with the smallest bias observed between the GCFD and machine learning products. Third, regarding temporal trends, GPP and RECO exhibit consistent patterns, while NEE shows an opposite trend. On the global scale, GPP values exhibit an overall increasing trend, with the tropics experiencing the most significant increase. Conversely, the polar regions display the lowest GPP values. Fourth, vegetation-related variables such as FAPAR and LAI play crucial roles in influencing the spatial and temporal variations in carbon fluxes. The temporal trends and spatial distribution of carbon fluxes align with the characteristics of FAPAR and LAI.

**Author Contributions:** W.S. (Wei Shangguan) conceived the research and secured funding for the research. W.S. (Wei Shangguan) and Z.X. designed the experiment. Z.X., W.S. (Wei Shangguan), Q.L., X.L. (Xingjie Lu) and Y.Z. performed the analyses. Z.X. wrote the first draft of the manuscript. Z.X. and W.S. (Wei Shangguan) conducted the research. W.S. (Wei Shangguan), V.N., X.L. (Xueyan Li), L.L., F.H., W.S. (Wenye Sun) and H.Y. reviewed and edited the manuscript before submission. All authors have read and agreed to the published version of the manuscript.

**Funding:** The study was partially supported by the National Natural Science Foundation of China (42088101, 42375144, U1811464, 41975122, 42075160, 4227515 and 42205149), the Guangdong Basic and Applied Basic Research Foundation (2021B0301030007, 2021A1515110215 and 2023A1515011996), and the Innovation Group Project of Southern Marine Science and Engineering Guangdong Laboratory (Zhuhai) (311020008).

**Data Availability Statement:** FLUXNET2015 and FLUXNET-CH4 are available at https://fluxnet.org/ (accessed on 7 June 2023). Drought-2018 is available at https://www.icos-cp.eu/data-products/YVR0-4898 (accessed on 7 June 2023). The GCFD can be accessed at https://doi.org/10.11888/Terre.tpdc.300009 (accessed on 21 June 2023). The FLUXCOM product is available at http://fluxcom.org/ (accessed on 21 June 2023). Products by Zeng et al. are available at https://doi.org/10.17595/20200227.001 (accessed on 21 June 2023). TRENDY model data are available at https://blogs.exeter.ac.uk/trendy/ (accessed on 21 June 2023). The product by Yuan et al. is available at https://doi.org/10.6084/m9.figshare.8942336.v3 (accessed on 21 June 2023). The GLASS product is available at http://www.glass.umd.edu/Download.html (accessed on 21 June 2023).

**Acknowledgments:** In situ carbon flux data used were provided by the FLUXNET community and ICOS team. We acknowledge FLUXCOM, Jiye Zeng, the TRENDY team, Wenping Yuan, and the GLASS team for providing upscaled carbon flux data.

**Conflicts of Interest:** The authors declare no conflict of interest.

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
