# Peer review of "Assessing the Reliability of Global Carbon Flux Dataset Compared to Existing Datasets and Their Spatiotemporal Characteristics"

_climate, doi:10.3390/cli11100205_

Round 1
Reviewer 1 Report
The work compared GCFD with existing data products, including two machine learning products: FLUXCOM and NIES, two ecosystem model products: TRENDY and EC-LUE, and one remote sensing product (Global Land Surface Satellite), at both site and global scales. The study is important in terms of giving the overview of data products reliability and biases over different region and is well written. I suggest following points to improve its quality before publication.
1. Can you mention about RECO and NEE also in the abstract along with GPP?
2. Third para of Introduction is reporting various findings but can be improved with a connection between the sentences.
3. Is there any dataset available with Carnegie–Ames Stanford Approach (CASA) too? If so, mention that in Introduction.
4. Kindly provide reference to equations mentioned in section 2.5.
5. Was there any quality check of data involved for all the different datasets? Any condition to discard any set of data below/beyond any value or due to occurrence of any weather event? Please mention it in the methodology.
6. I suggest commenting on the trends over China, India and other countries with fast-paced development with respect to Figure 3 and Figure 5. You may cite as reported by other studies:
https://doi.org/10.3390/land11040538; https://doi.org/10.1186/s40663-020-00229-0; https://doi.org/10.1371/journal.pone.0247165
7. Figure 7 is difficult to conclude. You may change the line diagrams to scatter plot.
8. Kindly mention in your manuscript, how you normalised the data shown in Figure 8.
9. I suggest the authors to write pointwise conclusion.
Author Response
The work compared GCFD with existing data products, including two machine learning products: FLUXCOM and NIES, two ecosystem model products: TRENDY and EC-LUE, and one remote sensing product (Global Land Surface Satellite), at both site and global scales. The study is important in terms of giving the overview of data products reliability and biases over different region and is well written. I suggest following points to improve its quality before publication.
Response: Thanks for your positive comments.
- Can you mention about RECO and NEE also in the abstract along with GPP?
Response: We have added contents about RECO and NEE in the abstract.
Modification: When using site observations as the benchmark, Gross Primary Production (GPP), Respiration of ECOsystem (RECO) and Net Ecosystem Exchange of machine learning products exhibits a higher R2 (ranging from 0.57 to 0.85, 0.53-0.79 and 0.31-0.70, respectively) compared to model products and remote sensing products.
- Third para of Introduction is reporting various findings but can be improved with a connection between the sentences.
Response: We have added some connections between sentences.
- Is there any dataset available with Carnegie–Ames Stanford Approach (CASA) too? If so, mention that in Introduction.
Response: Yes, we added it in the introduction.
Modification: Zhou et al. used the biogeochemical model Carnegie Ames Stanford Approach (CASA) to develop a dataset of GPP, RECO and NEE for North America [57].
- Kindly provide reference to equations mentioned in section 2.5.
Response: We have added reference to the equations.
- Was there any quality check of data involved for all the different datasets? Any condition to discard any set of data below/beyond any value or due to occurrence of any weather event? Please mention it in the methodology.
Response: No additional quality check was done as these datasets have been quality controlled by the data developer.
Modification: No additional quality check was done, and all data are used for the comparison.
- I suggest commenting on the trends over China, India and other countries with fast-paced development with respect to Figure 3 and Figure 5. You may cite as reported by other studies:
https://doi.org/10.3390/land11040538; https://doi.org/10.1186/s40663-020-00229-0; https://doi.org/10.1371/journal.pone.0247165
Response: We have added the content and references in section 3.2.
Modification: From Figure 3 and Figure 5, it is clear that in some countries with fast-paced development, such as China and India, the carbon fluxes showed obvious variations over seasons and trend s over years, which were driven by factors such as aerosol, precipitation and temperature [54-56]
- Figure 7 is difficult to conclude. You may change the line diagrams to scatter plot.
Response: This figure is about temporal trends, so we think line plots are more suitable for recognizing the trends.
- Kindly mention in your manuscript, how you normalised the data shown in Figure 8.
Response: Thanks for your advice. We have explained the way of normalizing in Figure 8.
Modification: Figure 8. Latitudinal mean distribution of carbon fluxes and FAPAR, LAI, air temperature (TA), and latent heat flux (LE), with data derived from the spatially averaged distribution of each variable from 1999 to June 2020, where GPP, RECO, and NEE are in g C m-2 d-1 and the remaining four predictors have been normalized (subtracted by the average and then divided by the standard deviation).
- I suggest the authors to write pointwise conclusion.
Response: We have made our conclusions point by point.
Modification: First, at the site scale, among the products, the GCFD product shows the closest agreement with the site observations in terms of predicted carbon fluxes. Second, the spatial distribution patterns of carbon fluxes are similar across all three product types, with the smallest bias observed between GCFD and machine learning products. Third, regarding the temporal trends, GPP and RECO exhibit consistent patterns, while NEE shows an opposite trend. On a global scale, GPP values exhibit an overall increasing trend, with the tropics experiencing the most significant increase. Conversely, the polar regions display the lowest GPP values. Fourth, Vegetation-related variables such as FAPAR and LAI play crucial roles in influencing the spatial and temporal variations of carbon fluxes. The temporal trends and spatial distribution of carbon fluxes align with the characteristics of FAPAR and LAI.
Reviewer 2 Report
Terrestrial carbon sinks support for achieving the goal of carbon neutrality, and it is crucial to clarify the carbon flux of terrestrial ecosystems. This manuscript systematically compares carbon fluxes estimated by multiple methods including in situ measurements, machine learning, satellite remote sensing, and model simulations. Detailed comments are as follows:
1. The manuscript uses carbon flux datasets from so many sources, and it is necessary to add a detailed table including data spatio-temporal resolution, estimation methods, etc. to increase readability.
2. The mutual comparison of carbon flux NEE in the manuscript is in the category of bottom-up methods, lacking the practicality of top-down atmospheric inversion, such as Carbon tracker and CAMS, it is recommended to add bottom-up and automatic Top-down NEE comparison.
3. In section 2.2, FLUXCOM has two products, RS and RS+METEO. Each product contains different climatological forces. Please specify which version you are using.
4. These carbon flux products usually have different spatio-temporal resolutions, and data processing is required to unify the resolutions, which is very important, but the author did not describe the data processing process in detail.
5. The author compared the newly developed GCFD with other machine learning products, but did not tell us in section 2.2 that the difference in the data production process of GCFD compared with FLUXCOM and NIES is that some improvements?
6. As far as I know, the latest TRENDY V11 contains 17 dynamic global vegetation models under 4 scenarios, and provides multiple carbon fluxes such as GPP, RECO and NEE. Why did the authors only use TREDNY GPP in section 2.3. There are serious problems with the description of TRENDY in this section.
7. In section 3.4, the author only analyzed the time series characteristics of several variables, and did not present the quantitative results of the correlation analysis and random forest prediction in section 2.6.
8. Intercomparison of time series requires listing specific differences in long-term trend and in interannual variability among different products.
No comments.
Author Response
Terrestrial carbon sinks support for achieving the goal of carbon neutrality, and it is crucial to clarify the carbon flux of terrestrial ecosystems. This manuscript systematically compares carbon fluxes estimated by multiple methods including in situ measurements, machine learning, satellite remote sensing, and model simulations. Detailed comments are as follows:
Response: Thanks for your positive comments.
- The manuscript uses carbon flux datasets from so many sources, and it is necessary to add a detailed table including data spatio-temporal resolution, estimation methods, etc. to increase readability.
Response: We have added Table 1 in section 2.
- The mutual comparison of carbon flux NEE in the manuscript is in the category of bottom-up methods, lacking the practicality of top-down atmospheric inversion, such as Carbon tracker and CAMS, it is recommended to add bottom-up and automatic Top-down NEE comparison.
Response: We agree that top-down atmospheric inversion can provide useful estimation of NEE with much uncertainty. However, as we majorly focus on NEE by top-down methods, we did not include top-down NEE in our paper.
- In section 2.2, FLUXCOM has two products, RS and RS+METEO. Each product contains different climatological forces. Please specify which version you are using.
Response: We used RS+METEO product and we have cleared it in section 2.2.
- These carbon flux products usually have different spatio-temporal resolutions, and data processing is required to unify the resolutions, which is very important, but the author did not describe the data processing process in detail.
Response: We have added the data processing method in section 2.5.
Modification: When different datasets are compared, the product with higher spatio-temporal resolution is processed according to the product with lower resolution using the averaging method
- The author compared the newly developed GCFD with other machine learning products, but did not tell us in section 2.2 that the difference in the data production process of GCFD compared with FLUXCOM and NIES is that some improvements?
Response: The improvements of GCFD are mainly its high spatial resolution and the deep learning method, which has been stated in “GCFD is a global land carbon fluxes dataset with a resolution of 1 km, generated using deep learning techniques trained with in-situ measurements from 280 stations”. We also added “Compared to FLUXCOM and NIES, GCFD is derived using deep learning with better performance than traditional machine learning and GCFD demonstrated more spatial details with higher resolution.”
- As far as I know, the latest TRENDY V11 contains 17 dynamic global vegetation models under 4 scenarios, and provides multiple carbon fluxes such as GPP, RECO and NEE. Why did the authors only use TREDNY GPP in section 2.3. There are serious problems with the description of TRENDY in this section.
Response: We used TRENDY v10 product and we can only access the GPP data during our data collection, and the server of TRENDY is not available now.
- In section 3.4, the author only analyzed the time series characteristics of several variables, and did not present the quantitative results of the correlation analysis and random forest prediction in section 2.6.
Response: The quantitative results of the correlation analysis and random forest prediction has been presented in our previous study at https://doi.org/10.3390/f14050913. To be clear, we modified the following sentence.
Modification: “To identify significant factors among the variables associated with carbon fluxes, this study employed two methods for attribution analysis, and the corresponding results are presented in the study by Shangguan et al [41]”
- Intercomparison of time series requires listing specific differences in long-term trend and in interannual variability among different products.
Response: We have discussed these differences among the products in section 4.1.
Modification: “Figure 11 compared the annual changes in carbon fluxes across different products. Among these products, the carbon fluxes obtained from GCFD predictions are relatively large, second only to the TRENDY predictions, and the trends align more closely with those derived from Zeng et al. The GPP and RECO values from GCFD exhibit an upward trend, accompanied by a decrease in NEE. In the GPP data for TRENDY products, there is also a noticeable upward trend in its year-to-year variation. This result indicates that the interannual variation of carbon fluxes predicted by GCFD is more pronounced compared to previous studies such as FLUXCOM and GLASS. Though the trends were similar among GCFD, Zeng et al. and TRENDY, their interannual variations are not exactly the same. For example, GCFD demonstrated an abrupt decrease of NEE around 2014, while the other two products did not.”
Reviewer 3 Report
Review of the manuscript “Assessing the reliability of Global Carbon Flux Dataset compared to existing datasets and its spatiotemporal characteristics” by Xiong et al.
The authors evaluate a product, the Global Carbon Fluxes Dataset (GCFD), by comparison to others commonly utilised, both based on remote sensing and modelling approaches. The assessment is performed both at site level and regional level; the spatio-temporal variations of the main variables are then evaluated.
In my opinion, a weak point in the manuscript is how the evaluation of the proposed dataset is performed. More attention should be given to the spatial resolution of each dataset and to the footprint area of the eddy covariance data considered. In addition, also the temporal resolution is important because not all products cover the same period. I suggest improving the discussion on this topic and, if possible, provide an integration between data at different resolutions to downscale/upscale the diverse dataset (obviously this effort might be proposed only for few sample cases).
In general, the manuscript falls in the aims of the journal and is well structured, however some points might be addressed prior its publication. Under a practical point of view, I would suggest utilising line numbers so that both the author and reviewer process is easier! Only minor, specific suggestions are given below.
Specific comments:
- Page 2, second paragraph, line 13: explain the acronym NCEP the first time it is mentioned.
- Page 3, Introduction, second last paragraph: change to “The study aimed at comparing…”
- Section 2: how did the authors treat the different spatial and temporal resolution of the global products utilised? The eddy-covariance measurements are referred to a specific footprint which is not comparable to the spatial resolution of global products….
- Section 2.2: are the 280 sites used by these authors the same described in Section 2.1? The authors should clarify and discuss this point.
- Section 2.6, page 5: specify at least the spatio-temporal resolution of the mentioned variables (LAI, fAPAR, etc). Are their characteristics easily comparable to those of the proposed dataset?
- Section 3.1: the first lines of the section can be moved to the methodological section where the authors should declare which product and at which spatio-temporal aggregation the comparison is performed.
- Section 3.1, second-last paragraph, four-last lines: this is a result section, maybe these lines should be moved to the discussion, where they are more appropriate. This comment should apply throughout the whole section. Moreover, repetitions should be avoided (e,g, the temporal resolution of some datasets is already provided in the Study data section.
- Page 5, end of section 2.6: add information about the spatio-temporal scales of the variables examined so that the reader is facilitated.
- Page 6, Figure 1: the determination coefficient of the second last scatter plot should be fixed, it cannot be negative and I am not sure that the measurement unit is correct, i.e., daily or monthly data as stated in the text? Moreover, the y-axis of all GPP scatter plots should be comparable (i.e., always set at the same amount).
- Section 3.2, lines 5-8: reformulate the sentence. The trends of the different variables should be evaluated also in relation to their mean values.
- Page 9, line 2: the climate regions should be introduced in the study data, even by means of a reference.
- Page 9, figure 4: this figure provides NEP (which is positive), not NEE. It can be done but the authors need to better specify this choice and be consistent with what stated in the text above.
- Page 10, Section 3.3: avoid including methodological and processing data description in a result section. Here and elsewhere in the result sections, also avoiding repetitions (e.g., the beginning of section 3.4).
- Page 11, end of section 3.3: better specifying “vegetation type”.
- Page 12, figure 7: the authors might split this figure into two sets to avoid confusion. What about the correlation analysis described in the methodological section? Maybe something should be added or its description can be removed.
- Page 15, figure 10: any ideas about the abrupt decrease detected in the GCFD NEE data during 2014?
- Page 15, section 4.3: maybe most of this section should be moved where data are described and indicated to validate the GCFD dataset at site level.
- Conclusions: reformulate the first paragraph of the section.
Author Response
Review of the manuscript “Assessing the reliability of Global Carbon Flux Dataset compared to existing datasets and its spatiotemporal characteristics” by Xiong et al.
The authors evaluate a product, the Global Carbon Fluxes Dataset (GCFD), by comparison to others commonly utilised, both based on remote sensing and modelling approaches. The assessment is performed both at site level and regional level; the spatio-temporal variations of the main variables are then evaluated.
Response: Thanks for your efforts in reviewing our manuscript.
In my opinion, a weak point in the manuscript is how the evaluation of the proposed dataset is performed. More attention should be given to the spatial resolution of each dataset and to the footprint area of the eddy covariance data considered. In addition, also the temporal resolution is important because not all products cover the same period. I suggest improving the discussion on this topic and, if possible, provide an integration between data at different resolutions to downscale/upscale the diverse dataset (obviously this effort might be proposed only for few sample cases).
Response:
For the week point of “how the evaluation of the proposed dataset is performed”. We have improved the description in the method section and added a new section 2.5.
We have summarized the information of different datasets including spatial and temporal resolution in Table 1. We have also added a sentence about the footprint area: “It should be noted that the resolution of GCFD (1 km) is closer to the footprint area of eddy covariance data, which makes it more comparable to in situ data than the other products with coarser resolution.”
We have described our method in handling the different resolutions: “When different datasets are compared, the product with higher spatio-temporal resolution is processed according to the product with lower resolution using the averaging method. Since the FLUXCOM data is available on a monthly basis, all datasets were processed to a monthly scale for accurate comparison.”
In general, the manuscript falls in the aims of the journal and is well structured, however some points might be addressed prior its publication. Under a practical point of view, I would suggest utilising line numbers so that both the author and reviewer process is easier! Only minor, specific suggestions are given below.
Response: Thanks for your suggestion. we have added line numbers in the right side. We addressed specific comments as follows.
Specific comments:
- Page 2, second paragraph, line 13: explain the acronym NCEP the first time it is mentioned.
Response: We have added the explanation of NCEP.
- Page 3, Introduction, second last paragraph: change to “The study aimed at comparing…”
Response: We have changed the expression.
- Section 2: how did the authors treat the different spatial and temporal resolution of the global products utilised? The eddy-covariance measurements are referred to a specific footprint which is not comparable to the spatial resolution of global products….
Response: When different products are compared, the data were processed according to the product with lower resolution, which we added in 2.4. When compared with the site observations, we selected the grids where the flux towers are located. For simplicity, we do not use any scaling method to compare the site data with the global products. We have addressed this limitation in the discussion: “Thirdly, the sites used for validation are limited in number and exhibit uneven spatial coverage, and there is a scale mismatch between the site-level observations and the grid-level estimations of carbon fluxes. These factors introduce uncertainties in the validation results of various carbon flux products.”
- Section 2.2: are the 280 sites used by these authors the same described in Section 2.1? The authors should clarify and discuss this point.
Response: The 280 sites in 2.2 are the same as the flux tower sites in section 2.1, which we have added in section 2.2.
Modification: GCFD is a global land carbon fluxes dataset with a resolution of 1 km, generated using deep learning techniques trained with in-situ measurements from 280 stations, which is the same as the flux tower sites in section 2.1.
- Section 2.6, page 5: specify at least the spatio-temporal resolution of the mentioned variables (LAI, fAPAR, etc). Are their characteristics easily comparable to those of the proposed datasets?
Response: We have added the information of resolution in section 2.7. We have compared their spatial and temporal variation and found that they are highly related.
Modification: FAPAR and LAI were obtained from Copernicus Global Land Service (CGLS) [51] with a spatial resolution of 1 km and temporal resolution of 10 days, while air temperature and latent heat flux were obtained from ERA5-Land [52] with a spatial resolution of 0.1 degree and temporal resolution of 3 hours.
- Section 3.1: the first lines of the section can be moved to the methodological section where the authors should declare which product and at which spatio-temporal aggregation the comparison is performed.
Response: We have added section 2.5 according to your advice.
- Section 3.1, second-last paragraph, four-last lines: this is a result section, maybe these lines should be moved to the discussion, where they are more appropriate. This comment should apply throughout the whole section. Moreover, repetitions should be avoided (e,g, the temporal resolution of some datasets is already provided in the Study data section.
Response: The second-last paragraph is about the result comparison between GCFD and GLASS. Maybe it is more suitable to be in section 3.1. And we have checked the repetitions carefully.
- Page 5, end of section 2.6: add information about the spatio-temporal scales of the variables examined so that the reader is facilitated.
Response: We have added this information according to your advice.
Modification: FAPAR and LAI were obtained from Copernicus Global Land Service (CGLS) [51] with a spatial resolution of 1 km and temporal resolution of 10 days, while air temperature and latent heat flux were obtained from ERA5-Land [52] with a spatial resolution of 0.1 degree and temporal resolution of 3 hours.
- Page 6, Figure 1: the determination coefficient of the second last scatter plot should be fixed, it cannot be negative and I am not sure that the measurement unit is correct, i.e., daily or monthly data as stated in the text? Moreover, the y-axis of all GPP scatter plots should be comparable (i.e., always set at the same amount).
Response: The negative R2 value came from the large difference between GPP from Yuan et al. And GPP from sites. R2 can be negative according to Equation 1, which indicates very poor performance. Here, we are showing daily carbon flux data of monthly mean, which was obtained by calculating the average value of all daily values in the same month. As you have advised, we have reset our y-axis.
- Section 3.2, lines 5-8: reformulate the sentence. The trends of the different variables should be evaluated also in relation to their mean values.
Response: We have changed our expression according to your advice.
Modification: The findings indicate that, among the three carbon flux variables, GPP and RECO demonstrated consistent trends over time, while NEE showed an opposite trend to GPP and RECO.
- Page 9, line 2: the climate regions should be introduced in the study data, even by means of a reference.
Response: We have added the introduce of climate zones in section 2.5.
Modification: Also, in addition to global scale, analysis has also been done in seven climate zones, including tropical, subtropical, continental, Mediterranean, oceanic, dry and polar climate, which was divided according to Koppen Climate Classification [50].
- Page 9, figure 4: this figure provides NEP (which is positive), not NEE. It can be done but the authors need to better specify this choice and be consistent with what stated in the text above.
Response: We provided NEP here due to the convenience of plotting. We have specified it in the caption of Figure 4.
Modification: We provided NEP instead of NEE here for the convenience of plotting, which is the opposite value of NEE.
- Page 10, Section 3.3: avoid including methodological and processing data description in a result section. Here and elsewhere in the result sections, also avoiding repetitions (e.g., the beginning of section 3.4).
Response: According to your advice, we have deleted methodological and processing data description and repetitions.
- Page 11, end of section 3.3: better specifying “vegetation type”.
Response: We have changed the words as you have advised.
- Page 12, figure 7: the authors might split this figure into two sets to avoid confusion. What about the correlation analysis described in the methodological section? Maybe something should be added or its description can be removed.
Response: We have split this figure into two sets. The correlation analysis has been done in our previous study at https://doi.org/10.3390/f14050913. To be clear, we modified the following sentence.
Modification: “To identify significant factors among the variables associated with carbon fluxes, this study employed two methods for attribution analysis, and the corresponding results are presented in the study by Shangguan et al [41]”
- Page 15, figure 10: any ideas about the abrupt decrease detected in the GCFD NEE data during 2014?
Response: We have described this phenomenon and try to explain it as follows.
Modification: “For example, GCFD demonstrated an abrupt decrease of NEE around 2014, while the other two products did not. This abrupt decrease may be due to the abrupt increase of global LAI and FPAR around 2014 as shown in Figure 8, when vegetation greening led to carbon sink.”
- Page 15, section 4.3: maybe most of this section should be moved where data are described and indicated to validate the GCFD dataset at site level.
Response: This section is not only about data description. The density of sites and its influence have also been discussed here, so we think it is appropriate to keep it in discussion.
- Conclusions: reformulate the first paragraph of the section.
Response: We have rewritten this paragraph as follows.
Modification:” In order to validate the accuracy of carbon flux predictions from GCFD generated by machine learning model and evaluate their capacity to accurately depict the actual spatial distribution of carbon fluxes, this study conducted a comparative analysis of three types of carbon flux products: remote sensing, ecosystem model, and machine learning products. The comparison was conducted at both the site scale and the global scale. Furthermore, this study examined the overall patterns of variation in carbon fluxes by analyzing their time series and spatial distribution. Additionally, an attribution analysis was performed, taking into account significant factors affecting carbon fluxes.”
Round 2
Reviewer 2 Report
NO
NO